# Emotional Distress in Portuguese Cancer Patients: The Use of the Emotion Thermometers (ET) Screening Tool

**DOI:** 10.3390/healthcare11192689

**Published:** 2023-10-06

**Authors:** Sónia Silva, Tiago Paredes, Ricardo João Teixeira, Tânia Brandão, Klára Dimitrovová, Diogo Marques, Joana Sousa, Monick Leal, Albina Dias, Carole Neves, Graciete Marques, Natália Amaral

**Affiliations:** 1Portuguese Cancer League (Central Branch), Rua Dr. António José de Almeida, nº 329—2º Sl 56, 3000-045 Coimbra, Portugal; tparedes@ligacontracancro.pt (T.P.); namaral@ligacontracancro.pt (N.A.); 2REACH—Mental Health Clinic, 4000-138 Porto, Portugal; ricardo@reach.com.pt; 3CINEICC, Faculty of Psychology and Educational Sciences, University of Coimbra, 3004-531 Coimbra, Portugal; 4William James Center for Research, ISPA—Instituto Universitário, 1149-041 Lisboa, Portugal; tbrandao@ispa.pt; 5Comprehensive Health Research Center, CHRC, NOVA University Lisboa, 1150-082 Lisboa, Portugal; klara.dimitrovova@moai-consulting.com; 6MOAI Consulting, 1350-346 Lisboa, Portugal; diogo.marques@moai-consulting.com (D.M.); joana.sousa@moai-consulting.com (J.S.); 7Portuguese Cancer League (North Branch), 4200-172 Porto, Portugal; monick.leal@ligacontracancro.pt; 8Portuguese Cancer League (South Branch), 1099-023 Lisboa, Portugal; albina.dias@ligacontracancro.pt; 9Portuguese Cancer League (Madeira Branch), 9050-023 Funchal, Portugal; cneves@ligacontracancro.pt; 10Portuguese Cancer League (Azores Branch), 9700-171 Angra do Heroísmo, Açores, Portugal; graciete.marques@ligacontracancro.pt

**Keywords:** distress thermometer, emotion thermometer, emotional distress, anxiety, depression, cancer

## Abstract

Cancer patients may experience significant distress. The “Emotion Thermometers” (ETs) are a short visual analogue scale used to screen patients for psychosocial risk. This study aimed to assess emotional distress in a large sample of cancer patients attending psychological services at an non-governmental organization (NGO), and to explore factors that may contribute to it. The ETs were answered by 899 cancer patients. They were, on average, 59.9 years old, the majority were female, had breast cancer, were under treatment or were disease-free survivors, and reported high levels of emotional distress, above the cut-off (≥5). A Generalized Linear Model was used to measure the association between the level of distress, age, gender, disease phase and 33 items of the problem list. Four items—sadness, depression, sleep and breathing—were found to be significantly related to a higher level of distress. Additionally, women and patients who were in the palliative phase also had significantly higher levels of distress. The results confirm the need for early emotional screening in cancer patients, as well as attending to the characteristics of each patient. Additionally, they highlight the utility of the ETs for the clinical practice, allowing to optimize the referral to specialized psychosocial services.

## 1. Introduction

In Portugal, a total of 60,467 new cancer cases were diagnosed in 2020, with the most prevalent being breast cancer for women and prostate cancer for men, followed by colorectal and lung cancers for both genders [1]. A cancer diagnosis can be a deeply overwhelming experience. Patients are likely to experience higher levels of anxiety, depression, distress, fear of cancer recurrence, sleep disturbance, body image disturbance and low self-esteem, posttraumatic stress disorders [2,3,4,5,6], among other difficulties at the interpersonal and social levels [7], such as loneliness, social isolation or the deterioration of social functioning and relationships, as well as spiritual and existentialism questioning [8]. Apart from that, patient and survivor’s experiences throughout the cancer journey involve multiple phases (pre-diagnosis, post-diagnosis, treatment, short-term after treatment, long-term survival or terminal illness and death), which entail different problems and demands [9].

In the field of oncology, the term “distress” encompasses a comprehensive range of unpleasant emotional experiences with psychological (e.g., cognitive, behavioural, emotional), social, and spiritual dimensions that exists on a continuum, spanning from typical feelings of vulnerability, sadness, and fear to more severe issues like depression, anxiety, panic, social isolation, and even existential and spiritual crises. These experiences have the potential to interfere with an individual’s ability to effectively cope with cancer, its physical symptoms, and its treatment [10,11]. Distress can be viewed as an integral part of the psychological adaptation process as individuals are faced with managing a cancer diagnosis, which inherently constitutes a highly stressful life event [12]. However, it can reach clinically significant levels among a substantial number of cancer patients [12]. Therefore, distress is considered the ‘sixth vital sign’ in psycho-oncology [13].

Thus, efficient screening for distress is crucial, which has led to the need for developing short screening tools. The Distress Thermometer (DT), developed by the National Comprehensive Cancer Network [14], was the first attempt to easily screen for distress in cancer patients. The distress experienced by cancer patients is not always evident for healthcare professionals and is likely to be underrecognized and undertreated, preventing patients from receiving adequate and timely psychological support. This has an impact on the quality of life, life satisfaction, and treatment adherence of the patients [15,16,17]. It may also influence treatment efficacy by affecting the immunosuppressive function [18] as well as increasing the risk of recurrence [19].

However, while the DT performs well in screening for distress, it is modestly accurate in screening for anxiety and depression [20]. Thus, in 2007, Alex Mitchell proposed a new multidomain extension of the DT, the Emotion Thermometers (ETs) tool to ensure that not only distress, but also anxiety, depression and anger could be easily detected by healthcare professionals. The ETs are a brief, visual analogue scale (VAS) and easy-to-use tool to identify cancer patients’ psychosocial distress. It includes five visual rating scales with different emotional states, namely: distress, anxiety, depression, anger, and need for help [21].

In a recent review concerning the use of ETs among individuals diagnosed with cancer, Harju et al. stated that overall ETs are a sensitive tool to screen for distress, as found by five validation and two exploratory studies, that recommended a cut-off equal to or greater than four [22]. In Portugal, the ETs were validated by Teixeira et al. using a sample of 147 cancer patients [23]. The authors found that the ETs have excellent discrimination and identified the optimal cut-off values for each ET. The ETs have been widely used not only with patients, but also caregivers [24] and patients with cardiovascular disease or epilepsy [25,26].

Different psychological theories attempt to explore the psychological processes, in terms of cognitive appraisals, coping mechanisms, and social factors that contribute to the experience of stress and distress in response to various life events or situations, such as a cancer diagnosis. For instance, cognitive appraisal plays a crucial role in shaping the emotional response and coping strategies adopted by individuals as they navigate their cancer journey [27]. When individuals receive a cancer diagnosis, they tend to undergo a cognitive appraisal process, evaluating the nature of the stressor and their ability to cope with it effectively, also accessing their available resources (such as financial, social support, and treatment access) [27,28], with a higher perceived threat being associated with increased distress. These appraisals also influence the coping strategies used by patients [29]. In the context of cancer, problem-focused coping strategies (such as seeking a diagnosis, gathering relevant information) often work as proactive steps being linked to higher well-being and less distress; conversely, coping strategies characterized by avoidance, such as self-distraction, have been strongly associated with adverse outcomes [30]. The active engagement in decision-making and the ability to take concrete actions to address the illness may contribute to a sense of empowerment and optimism, thus promoting emotional well-being during cancer survivorship.

Additionally, several factors have been identified in the literature as risk factors for developing distress, including sociodemographic and clinical factors, although with contradictory results. Younger age, female gender, low education, and lower socioeconomic level, as well as unemployment status, have been associated with higher levels of distress in the majority of studies [12,31,32,33,34,35,36,37,38]. On the other hand, in other studies, being married and living in urban locations have been found to be protective factors [33,37,39].

Concerning clinical variables, greater distress has been associated with higher severity and more advanced stages of the disease in some studies [12,34,40,41,42,43,44,45], but no relationship between such variables has been found in other investigations [37,46]. Patients with no evidence of disease were less likely to report high distress compared to patients with newly diagnosed cancer or in active treatment, in the study of Liu and colleagues [47]. The prevalence of distress also seems to vary with tumour location and type of treatment. Namely, patients diagnosed with digestive system malignancies, breast and genitourinary cancers, hematologic, lung and head and neck malignancies, as well as patients receiving chemo or radiotherapy, were found to have higher rates of clinical distress [12,48,49]. Again, these results were not consistently found, and the same occurs with time since diagnosis, which has not always been correlated with distress or psychological adjustment [37,45,50,51]. Methodological issues such as different sample sizes, tumour location, cancer stages, measures used to assess distress and different procedures may explain these inconsistencies.

A study has also identified financial concerns, worry, nervousness, difficulties in mobility, and sleep issues as the most frequently reported problems associated with distress, according to the ETs problem list [52]. A study conducted on 3724 cancer patients diagnosed across various tumour locations and with an average age of 58 years revealed that fatigue, sleep disturbances, and mobility issues were the most frequently reported [12]. Other studies have identified emotional problems (worry, nervousness, depression, sadness), insurance-/finance-related problems, and sleep issues as the most common problems reported by cancer patients [53].

This study aimed to screen emotional distress in a sample of Portuguese cancer patients while initiating psychological intervention in an NGO. More specifically, the objectives were:(1)To characterize cancer patients attending the Psycho-Oncology services of the Portuguese Cancer League, in terms of sociodemographic and clinical variables;(2)To identify and differentiate cancer patients’ emotional distress, using the Portuguese cutoff points of the ETs screening tool;(3)To further study the relationships among distress, age, gender, clinical variables, and 33 items of the NCCN Distress Thermometer Problem List;(4)To identify the variables that, in the present sample of cancer patients, contribute the most to the distress levels.

We hypothesize that cancer patients attending the Psycho-Oncology Units would experience high levels of emotional distress, since they have requested or were referred by a health care professional to psychological help in this NGO. We also expect that socio-demographic and clinical variables, as well as problems of the ET screening tool, would be associated with distress.

## 2. Materials and Methods

This was a cross-sectional study that employed a questionnaire to adult patients at their first consultation at the Psycho-Oncology Units of the Portuguese Cancer League (NGO). A sample was collected across all five geographical Portuguese regions, between February 2020 and February 2023. During this period, 899 out of 1450 adult cancer patients who were admitted in these units answered the questionnaire (response rate of 62%). Patients aged 18 years old or more and diagnosed with any type of cancer and in any phase of the disease were included. Patients unable to understand Portuguese or with a severe physical or mental condition as well as patients who started the psychological intervention via telephone were excluded.

Approximately half of these patients were referred by a health professional, due to their potential increased risk for psychological distress, while the other half seek this support without a formal referral. Therefore, the sample included in this study corresponds to a convenience sample of patients who were attending for the first time the specialized psychological services at this NGO during the study period.

### 2.1. Measures

The protocol included demographic (age, gender) and clinical questions (disease phase and type of cancer), the ETs, as well as the original version of the problem list with 33 items [54]. Additionally, the researchers added five items within the section “other problems” to ascertain further concerns: economic, legal, communication with health professionals, body image, and information about the disease (Appendix A).

Concerning the Portuguese version of the ETs, it consists of five visual-analogue thermometers, including the following emotional domains: distress, anxiety, depression, anger, and need for help. Patients indicated the amount of emotional distress they felt in the past week including today, on the five ETs, by selecting a number from 0 (no distress) to 10 (extreme distress). A higher score indicates the greatest levels for each emotional domain.

The Cronbach’s alpha for the Portuguese sample was 0.93 [16]. The following optimal cutoffs were found with the Portuguese sample: distress thermometer, 4v5 (until 4 and 5 or more); anxiety thermometer, 5v6 (until 5 and 6 or more); depression thermometer, 4v5 (until 4 and 5 or more); anger thermometer, 4v5 (until 4 and 5 or more); help thermometer, 3v4 (until 3 and 4 or more) [16].

The questionnaires were self-administered while patients were in the waiting room for the first Psycho-Oncology consultation, but in some cases, they were filled out with the help of a psychologist. The list of problems required simply a yes/no answer.

### 2.2. Statistical Analyses

Descriptive statistics were used to characterize the sample and to identify items on the problem list with high prevalence. Emotional distress was described in terms of frequency and mean (SD).

#### Generalized Linear Model

We used the level of distress, measured on a continuous scale (from 0 to 10) as a dependent variable, and the items of the problem list as independent variables, to determine which of them were the most distressing. Age (years, continuous), gender (dichotomous), and disease phase (nominal: diagnosis, treatment, relapse, survival, palliative) were also included as independent variables, as they have previously been shown to demonstrate a relationship with distress [38]. In a first exploratory analysis, a separate model was performed for each independent variable, to ascertain their relationship with distress (Appendix A). Only the statistically significant covariates, considering a 95% confidence interval, were then considered entered in the final adjusted generalized linear model (GLM).

Specifically, a GLM with gamma distribution and log link was used [55]. For the simplicity of interpretation, results are presented as marginal effects (i.e., the incremental effect in the dependent variable, resulting from a change in the covariate). Additionally, the predicted margins (i.e., adjusted means) for the statistically significant covariates, considering a 95% confidence interval, are presented. All analyses were performed with Stata version 16.1.

## 3. Results

### Descriptive Analysis

A total of 899 patients (79.2% female) with a mean age of 59.9 years (SD = 12.6 years) answered the questionnaire. Most patients were in the treatment phase (48.7%) or in the survival phase (32.4%), while 8.9%, 5.9% and 4.1% were in the diagnosis, relapse, and palliative phase, respectively. The most frequent cancer type was breast (50.7%), followed by colorectal (8.2%), lung (5.6%) and lymphoma (4.7%) (Table 1).

The distribution of patients reporting emotional distress (0 to 10) in each of the five ETs, as well as the percentage of patients with emotional distress over cut-offs, are presented in Figure 1.

The results show a left-skewed distribution for all ETs, except for anger, with most patients reporting high levels of emotional distress. For the distress thermometer, 87% of patients are above the cut-off (≥5) (Figure 1). The most frequently reported problems were worry (91.9%), sadness (83.9%), fears (80.6%), nervousness (78.4%), fatigue (75.2%), sleep (61.7%), and pain (59.0%) (Figure 2).

Results from the GLM show that there are four items in the problem list—sadness, depression, sleep and breathing—that are significantly related to a higher level of distress. Specifically, marginal effects demonstrate that, on average, patients who report sadness have a level of distress 1.42 points higher than patients who do not report sadness, when adjusting for all other covariates (marginal effect = 1.42, *p* < 0.01). Similarly, patients who report depression have a level of distress 1.13 points higher than patients who do not (marginal effect = 1.13, *p* < 0.01). Patients with sleeping problems and breathing problems also have a higher level of distress, with a marginal effect of 0.55 (*p* < 0.01) and 0.63 (*p* < 0.01), respectively. In addition to the items of the problem list, being female also contributes to a higher level of distress (0.59 points higher (*p* < 0.01) as compared to male patients. Finally, patients currently in the palliative phase have a level of distress 1.33 points higher than patients in the treatment phase (*p* < 0.01) (Table 2). The full results are available in Appendix A.

The predicted margins for the abovementioned covariates are shown in Figure 3.

These adjusted means represent the average level of distress within each category. In other words, they represent the average level of distress for female patients and male patients within the observed sample, adjusted for all the other covariates. Note that for the disease phase, only the level of distress of patients in the palliative phase in comparison to the treatment phase showed a statistically significant difference.

## 4. Discussion

The aim of this study was twofold. First, it aimed to identify the levels of distress experienced by cancer patients seeking psychological support at an NGO. Second, it aimed to identify the factors that most contribute to this distress to better understand and address these problems in the psychological support provided to these individuals. The evaluation of the distress was carried out using the ETs, a brief and easy-to-use tool to identify cancer patients’ distress [21].

Overall, the results showed, as expected, that most of the patients experienced high levels of distress, with 87% of patients being above the cut-off (≥5) proposed by Teixeira et al. [23]. Thus, it confirms the results of previous studies showing that cancer patients tend to experience a wide range of negative outcomes [2,3,4,5,6]. In a previous review [22], the percentage of individuals experiencing clinically significant distress (with a cut-off ≥4) ranged from 13% [56] to 88.5% [21]. The prevalence in our study is higher than most previous studies (except for [21]), what was expected considering that participants were seeking psychological support (either by self-initiative or by referral from health professionals), which effectively highlights the patients’ need for specialized help. It is also important to acknowledge that part of the data for this study was collected during the COVID-19 pandemic, which may have exacerbated the distress experienced by cancer patients, including those seeking psychological support in a cancer NGO. This is consistent with previous findings, in line with appraisal theory, which postulates that the psychological well-being and distress of cancer patients were greatly affected by how they evaluated and interpreted the threat posed by the COVID-19 outbreak, as well as their perceived capabilities to cope with the unique challenges brought on by the pandemic [57,58,59]. The heightened distress observed in cancer patients during this time may be attributed to the increased perceived threat of contracting the virus and the uncertainties surrounding treatment access and healthcare services, which might have amplified feelings of vulnerability and a reduced sense of control. As showed in a systematic review [60], the COVID-19 pandemic has posed several challenges to all aspects of cancer diagnosis and treatment, including to early detection, disease progression, initiation of, interruption or stop in treatments, and to overall and disease-free survival, which may have contributed to an increased risk for emotional distress. Furthermore, disruptions in social support networks and limited opportunities for meaning-making due to the pandemic might have further contributed to their emotional struggles.

Since participants who seek psychological support are those who, supposedly, are more aware of their distress and more willing to acknowledge and discuss their feelings, as evidenced by some studies [61,62], it is also important to stress that many other cancer patients may be experiencing significant distress without receiving any specialized emotional support. Therefore, it is important to conduct studies that address these patients, as well as develop strategies that allow their identification and proper referral for psychosocial services.

There are multiple causes of distress, and they may differ greatly between people. Assessing these causes is extremely important since it allows the implementation of a treatment plan according to the individual’s needs. Based on the list of problems [54], we found that the main risk factors for experiencing elevated distress among cancer patients were mostly physical (sleep disturbance and breathing difficulties) and emotional problems (sadness and depression). These results are in line with previous studies [12,37,52,53]. Sleep disturbances and breathing difficulties are prevalent symptoms in cancer patients, particularly in those with metastatic cancer, and are often interrelated [63,64]. These symptoms can contribute to the magnification of the distress experienced by cancer patients (for instance, by impacting daily physical activity). Sadness and depression are also common experiences among cancer patients. This pattern is found in most studies, as highlighted by some authors [65]. Appraisal and coping theories provide valuable explanatory frameworks for understanding the prevalence of sadness and depression among cancer patients. These theories emphasize how cancer patients’ cognitive appraisals of their diagnosis, treatment journey, and available coping resources play a pivotal role in shaping their emotional responses, making sadness and depression common and understandable reactions to the profound challenges they encounter [27,28]. Additionally, being a woman and in a palliative phase of the disease were also risk factors for experiencing high distress, which also confirms previous studies [34,38].

Past literature [66] has reported that practical, family, and religious problems are less likely to cause heightened distress among cancer patients. Similarly, our findings also suggest that these factors may not be significant contributors to distress in this population. However, as pointed out by Jewett et al. [67], the problem list included in the ETs may not accurately identify the concerns that are most strongly associated with high levels of distress in cancer patients. As we found, the most prevalent problems identified were not those that predicted higher levels of distress (e.g., worry, fears, fatigue, pain), with the exceptions being sadness and sleep. In contrast, while breathing difficulties were not among the most prevalent problems, our analysis revealed a significant association between these symptoms and higher levels of distress in cancer patients. Yet, the problem list can be useful for a quick screening of the patients’ problems, allowing clinicians to save time when evaluating patients and also target referrals to a multidisciplinary team. However, it does not exclude the need for a careful and rigorous evaluation of patients’ difficulties and other sources of distress in the clinical interview.

Our review of the literature [68] and clinical experience led us to include additional items, such as body image concerns, communication with health professionals or legal problems, in the problem list beyond the original set (some of these items are also included in the 2023 version of the list problems proposed by the NCCN). Despite their significant association with distress in the unadjusted analysis (Appendix A), we did not find a significant association between these added items and levels of distress in cancer patients in our adjusted model. While these items may be clinically relevant and worth considering in patient assessments (e.g., concerns about body image were a prevalent problem), our findings suggest that they may not be as strongly linked to distress as other emotional and physical problems.

Nevertheless, since emotional problems and distress may have an overlapping definition as “a multifactorial unpleasant emotional experience” [69], we performed an additional GLM adjusted model to explore the relationship between distress and the problem list items, without emotional items (i.e., when the five emotional problems are excluded from the analysis). We found a significant association between distress and problems with work/school, dealing with children, loss of faith, eating problems, and concerns about body image (in addition to sleep and breathing). Results can be found in Appendix A. Taking into account these results, it is also important to consider and analyse eventual mediation processes by the emotional problems/items.

### 4.1. Implications for Clinical Practice

We believe this study has important clinical implications for all professionals working with cancer patients. First, by suggesting that a considerable number of cancer patients manifest significant emotional distress throughout the cancer journey, it draws attention to the need for early screening for distress, and its monitorization in each phase of the disease. Second, it identifies the most common sources of distress and patients who may be at greater risk of developing high levels of distress, in terms of sociodemographic and clinical characteristics, helping healthcare providers to maket appropriate referrals to professionals or interventions that may be most effective in addressing cancer patients’ specific needs, as well as mental health professionals to tailor interventions accordantly. Third, the information collected in ETs and the problem list can also be useful to health professionals to adjust their speech and general communication with the patient. They also may use this information to psychoeducate patients and their relatives/caregivers about common causes of distress, empowering them to seek support/manage distress.

### 4.2. Limitations and Future Research

This study presents some limitations that should be acknowledged. This is a cross-sectional study, which limits conclusions regarding the causality between the identified problems and the experienced distress. Additionally, due to this specific design, we cannot fully capture the trajectory of distress experienced by cancer patients throughout the course of their disease. Also, participants were asked to report their distress level only in the previous week, limiting conclusions regarding their levels of distress at different moments. Longitudinal studies are needed to better understand trajectories of distress and related factors in the different phases of the disease.

The convenience sampling is another limitation. Although patients were recruited at a national level, some sampling bias may be present in terms of sample representativity, limiting the generalization of results.

On the other hand, the heterogeneity of the participants also represents another sampling bias and, again, with limitations for the generalization of results, for instance, for different types of cancer, phases of disease or treatments. In the present study, the specific effects of the different socio-demographic and clinical variables on distress were examined in order to better understand some of the individual differences on emotional adjustment. Even so, some caution is required in the interpretation of results and of conclusions drawn. Future research should focus on more homogeneous groups and considering variables that may be specific to different types of cancer, phases of disease and treatments.

Moreover, participants of this study were seeking psychological support, and the data were collected during the COVID-19 pandemic, which may have contributed to the higher levels of distress experienced by these participants. Future studies should also include patients not seeking psychological support to examine their levels of distress and their list of problems.

Finally, patients may underreport or overreport their distress due to several reasons (e.g., social desirability). Also, additional problems that were included in the latest version of the NCCN’s DT were not included in the present study (e.g., ability to have children, access to medicine); thus, future studies should explore the role of these new problems in explaining individual differences in the distress experienced.

## 5. Conclusions

In conclusion, the results of our study confirm the need for early emotional screening of cancer patients’ distress, as well as the relevance of attending to the characteristics of each patient. They also highlight the utility of the ETs for clinical practice, allowing for the optimization of the referral to specialized psychosocial services and better adjust the intervention according to the distress levels and reported needs.

## Figures and Tables

**Figure 1 healthcare-11-02689-f001:**
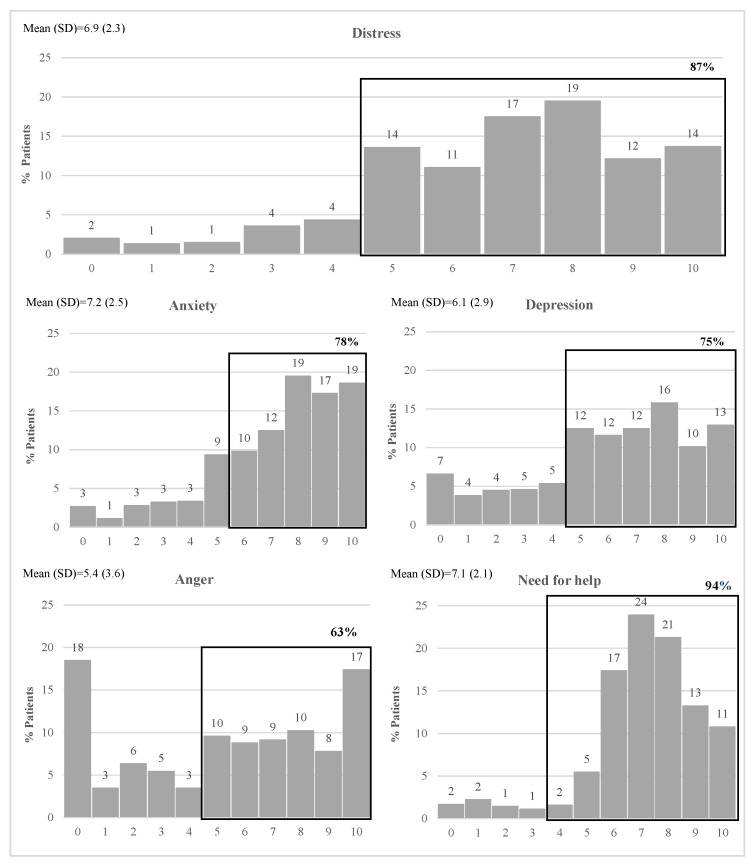
Prevalence of emotional distress, considering the five Emotion Thermometers: distress, anxiety, depression, anger, and need for help.

**Figure 2 healthcare-11-02689-f002:**
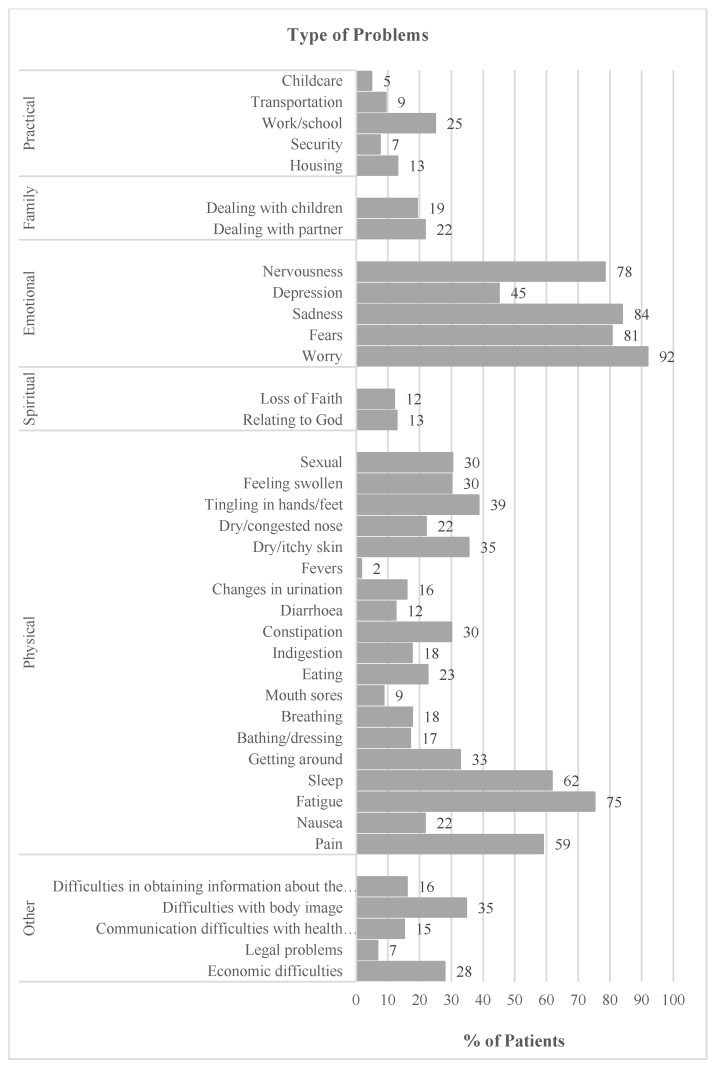
Prevalence of each problem type.

**Figure 3 healthcare-11-02689-f003:**
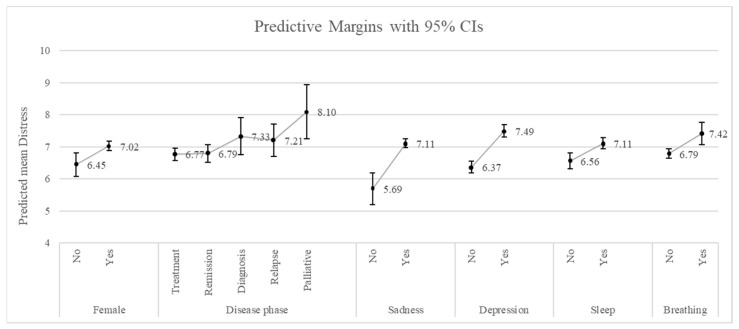
Predicted margins of the generalized linear model, for the relationship between covariates and distress.

**Table 1 healthcare-11-02689-t001:** Patients’ demographics and clinical characteristics.

Characteristic	Patients n = 899 N (%)
**Demographics**	
Age [mean (SD)]	59.9 (12.6)
Female	712 (79.2)
**Disease phase**	
Treatment	438 (48.7)
Survival	291 (32.4)
Diagnosis	80 (8.9)
Relapse	53 (5.9)
Palliative	37 (4.1)
**Type of Cancer**	
Breast	456 (50.7)
Colorectal	74 (8.2)
Lung	50 (5.6)
Lymphoma	42 (4.7)
Brain	27 (3.0)
Stomach	24 (2.7)
Thyroid	22 (2.5)
Pancreatic	17 (1.9)
Ovaries	16 (1.8)
Multiple myeloma	16 (1.8)
Prostate	15 (1.7)
Leukaemia	13 (1.5)
Cervical	12 (1.3)
Other	115 (12.8)

**Table 2 healthcare-11-02689-t002:** Generalized linear model results for the relationship between patient characteristics and problem list, and distress.

	Distress	
	Marginal Effect	Std. Err.
**Demographics**		
Female	0.57 ***	0.21
**Disease phase**		
Treatment	Ref.	
Survival	0.02	0.18
Diagnosis	0.56	0.31
Relapse	0.44	0.28
Palliative	1.33 ***	0.45
**Emotional**		
Sadness	1.42 ***	0.27
Depression	1.13 ***	0.15
**Physical**		
Sleep	0.55 ***	0.17
Breathing	0.63 ***	0.20

Note: Only the results for the statistically significant covariates, considering a 95% confidence interval, are presented; *** *p* < 0.01.

## Data Availability

Part of the data can be found in https://www.ligacontracancro.pt/sofrimentoemocional/.

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
