# Peer review of "Emotional Distress in Portuguese Cancer Patients: The Use of the Emotion Thermometers (ET) Screening Tool"

_healthcare, 2023, doi:10.3390/healthcare11192689_

Round 1

Reviewer 1 Report

Methodological Biases exist (selection biases, mainly)

The Authors must see my remarks

It requires MODERATE revision

Author Response

First, we'd like to thank you for all your comments and suggestions, which were very valuable for the improvement of the quality of the present manuscript!

All comments have been considered by our team and we have made all the efforts to address each of them.

Below, we have answered to each comment individually, and you can see the changes made directly in the text.

In the hope that the changes made meet your revisions, we remain open to adjusting any aspects that you deem necessary.

  1. “Methodological Biases exist (selection biases, mainly)”
    Response: The main aim of the present study was to have a description/characterization of the patients attending consultations at the Psycho-Oncology Unit of Portuguese Cancer League*, in terms of sociodemographic and clinical variables, emotional distress and sources of distress (problem list).
    It was, therefore, a convenience sample: patients were not previously selected for this study and no sample size was determined. However, inclusion and exclusion criteria were considered and are now included in the main text – section materials and methods (text underlined in yellow).
    This methodology leads to a sampling bias, due to the convenience recruitment and sample heterogeneity, which mask important individual differences and limit the generalization of results. However, to overcome these limitations, we perform a statistical control of this bias, by analysing the specific effects of sociodemographic and clinical variables on distress to better understand some of the individual differences on emotional adjustment.
    These limitations were included in the manuscript and some caution in the interpretation of results and conclusions drawn were highlighted.

*The Psycho-Oncology Units of the Portuguese Cancer League were created in 2009 and are now represented in 54 different locations of the country. They have the collaboration of 55 clinical psychologists trained in the field of psycho-oncology, who attend around 2000 cancer patients/caregivers, annually.

As part of the assistance work developed by these Units, it is a usual procedure to apply the Emotional Thermometers to all patients while attending for the first Psycho-Oncology consult, to screen for emotional distress. In the scope of this work, we planned to carry out this study which resulted in the submitted manuscript. The study is still ongoing.

  1. “The Authors must see my remarks”
    Response: all the remarks have been considered and all the changes made are underlined in the text in yellow colour (the text underlined in pink and green corresponds to changes suggested by other reviewers).

Specifically:

  • Model´s used: “Additionally, women and those who were in the palliative phase also had significantly higher levels of Distress.” (p.1)
    Response: The paragraph concerning the GLM was changed in order to clarify the variables included in the model.
  • Reference: “Distress Thermometer (DT), developed by the National Comprehensive Cancer Network” (p.2)
    Response: The reference was added.
  • “Do not state explanations “The GLM is a flexible generalization of linear regression..”” (p.3)
    Response: The explanation of the GLM was removed.
  • “Tables should be more useful”
    Response: We have chosen to keep the figures as we consider it is important for a quick visualization of the descriptive data and taking in consideration the other reviewers' remarks.
  • “Avoid stating personal opinions or conclusions” (p.9)
    Response: the text was adjusted to avoid any unwanted personal opinion.
  • “Implications for clinical practice – remove all paragraph” (p.9)
    Response: Authors consider this section important due to its contributions to other professionals working in the field of Psycho-Oncology. We also had good feedback from other reviewers, so the paragraph concerning the implications for clinical practice were maintained with some adjustments.

Reviewer 2 Report

Comments on the article “Emotional distress in Portuguese cancer patients: the use of the Emotion Thermometers” submitted to Healthcare

Thank you for inviting me to review this interesting manuscript. It is written in a neat way. It provides important findings related to cancer patients’ distress.

Below you will find my comments and tips on how to improve the manuscript.

1.     The title and abstract cover the main aspect of the work. They include adequate names of the examined variables. The abstract briefly and accurately describes the results of the study.

2.     The introduction is written quite clearly and logically, but not comprehensive. It provides definitions and results of previous research on the analyzed variables. Factors related to distress have been shown, but there are no conceptualizations of distress. I recommend deepening the theoretical background by presenting conceptualizations of distress, especially emotional one, in the context of cancer, along with psychological theories that explain why and how different factors are stressful or painful.

3.     I did not find any research questions and hypotheses. As a result, it is difficult to assess whether the method is adequate to the objectives/hypotheses of the study.

4.     The statistical analysis seem correct.

5.     Tables and figures are clear, legible and free from unnecessary modification.

6.     Since the theoretical background is insufficient, the discussion part is a bit superficial, but I appreciate the implications for clinical practice.

Author Response

First, we'd like to thank you for all your comments and suggestions, which were very valuable for the improvement of the quality of the present manuscript!

All comments have been considered by our team and we have made all the efforts to address each of them.

Below, we have answered to each comment individually, and you can see the changes made directly in the text.

In the hope that the changes made meet your revisions, we remain open to adjusting any aspects that you deem necessary.

  1. The title and abstract cover the main aspect of the work. They include adequate names of the examined variables. The abstract briefly and accurately describes the results of the study.
    Response: We appreciate reviewer's remarks. Title and abstract were slightly changed as suggested by other reviewer.

  2. The introduction is written quite clearly and logically, but not comprehensive. It provides definitions and results of previous research on the analysed variables. Factors related to distress have been shown, but there are no conceptualizations of distress. I recommend deepening the theoretical background by presenting conceptualizations of distress, especially emotional one, in the context of cancer, along with psychological theories that explain why and how different factors are stressful or painful.
    Response: The introduction section was reformulated following the reviewer recommendations and the changes made are underlined in pink colour (the text underlined in yellow and green corresponds to changes suggested by other reviewers).

  3. I did not find any research questions and hypotheses. As a result, it is difficult to assess whether the method is adequate to the objectives/hypotheses of the study.
    Response: General and specific aims and hypotheses were included in the main text.
  4. The statistical analysis seem correct.
    Response: OK

  5. Tables and figures are clear, legible and free from unnecessary modification.
    Response: We appreciate your feedback.

  6. Since the theoretical background is insufficient, the discussion part is a bit superficial, but I appreciate the implications for clinical practice.
    Response: Some changes in the discussion section were made to integrate the theoretical background deepened in the introduction. 

Reviewer 3 Report

In tittle, Emotion Thermometers is ambiguous.

If possible, give more details in the abstract (method and results)

Remove the: Factors related to distress (But not the text.)

If possible, the introduction should be more concise.

The sample size and sampling is not clear.

The validity, reliability and scoring of the questionnaire should be stated.

Implications for clinical practice AND Limitations and future research should be more concise.

During the Covid pandemic , the psychological and screening problems of cancer patients have increased, Impact of the COVID-19 Pandemic on Colorectal Cancer Screening: a Systematic Review  be considered in the discussion.

Author Response

First, we'd like to thank you for all your comments and suggestions, which were very valuable for the improvement of the quality of the present manuscript!

All comments have been considered by our team and we have made all the efforts to address each of them.

Below, we have answered to each comment individually, and you can see the changes made directly in the text.

In the hope that the changes made meet your revisions, we remain open to adjusting any aspects that you deem necessary.

  1. In tittle, Emotion Thermometers is ambiguous.
    Response: Title was changed, according to the reviewer comment. Changes/additions to the text are underlined in green (the text underlined in pink and yellow corresponds to changes suggested by other reviewers).

  2. If possible, give more details in the abstract (method and results)
    Response: Method and results have been slightly changed in order it to be more explicit. It is quite difficult to add more information given the word limit.

  3. Remove the: Factors related to distress (But not the text)
    Response: OK

  4. If possible, the introduction should be more concise.
    Response: Since the number of characters was less than desired and other reviewers asked to expand on certain aspects of the introduction, we opted to keep the topics mentioned, with the requested additions.

  5. The sample size and sampling is not clear.
    Response: Sample size and sampling were clarified in the “Materials and methods section”.

  6. The validity, reliability and scoring of the questionnaire should be stated.
    Response: Psychometric data and scoring of the questionnaire were stated in measures section.

  7. Implications for clinical practice AND Limitations and future research should be more concise.
    Response: Adjustments were done in the text.

  8. During the Covid pandemic, the psychological and screening problems of cancer patients have increased, Impact of the COVID-19 Pandemic on Colorectal Cancer Screening: a Systematic Review be considered in the discussion.
    Response: The Systematic Review mentioned was added in the text.

Round 2

Reviewer 2 Report

Comments on the article “Emotional distress in Portuguese cancer patients: the use of the Emotion Thermometers” submitted to Healthcare

The authors did a lot of work on the first draft of their submission, resulting in a better final manuscript. They improved the introduction and methodological section according to the reviewers' comments. They applied appraisal and coping theories for understanding emotional responses among cancer patients, which helped them better justify the study. I believe the manuscript is ready for publication. Congratulations!

Author Response

Dear Reviewer,
Thank you for your kind words! They are an important stimulus for future work.
Thank you and please accept our best regards!